# A Study on the Structure and Biomedical Application Characteristics of Phosphate Coatings on ZKX500 Magnesium Alloys

**DOI:** 10.3390/bioengineering9100542

**Published:** 2022-10-11

**Authors:** Ying-Ting Huang, Wen-Yu Wu, Fei-Yi Hung, Fa-Chuan Kuan, Kai-Lan Hsu, Wei-Ren Su, Chen-Wei Yen

**Affiliations:** 1Department of Materials Science and Engineering, National Cheng Kung University, Tainan 701, Taiwan; 2Orthopaedic Department, College of Medicine, National Cheng Kung University, Tainan 701, Taiwan; 3Ting Sin Co., Ltd., Tainan 717, Taiwan

**Keywords:** magnesium alloy, anodic oxidation, degradation, mechanical properties, implant, animal experiment

## Abstract

Magnesium-matrix implants can be detected by X-ray, making post-operative monitoring easier. Since the density and mechanical properties of Mg alloys are similar to those of human bones, the stress-shielding effect can be avoided, accelerating the recovery and regeneration of bone tissues. Additionally, Mg biodegradability shields patients from the infection risk and medical financial burden of needing another surgery. However, the major challenge for magnesium-matrix implants is the rapid degradation rate, which necessitates surface treatment. In this study, the ZKX500 Mg alloy was used, and a non-toxic and eco-friendly anodic oxidation method was adopted to improve corrosion resistance. The results indicate that the anodic coating mainly consisted of magnesium phosphate. After anodic oxidation, the specimen surface developed a coating and an ion-exchanged layer that could slow down the degradation and help maintain the mechanical properties. The results of the tensile and impact tests reveal that after being immersed in SBF for 28 days, the anodic oxidation-treated specimens maintained good strength, ductility, and toughness. Anodic coating provides an excellent surface for cell attachment and growth. In the animal experiment, the anodic oxidation-treated magnesium bone screw used had no adverse effect and could support the injured part for at least 3 months.

## 1. Introduction

Magnesium is a potential implant with a density of about 1.74 g/cm^3^ and an elastic modulus of about 40 GPa [1]. Its properties are more similar to those of human bones than other metals, thus, avoiding the stress-shielding effect [2]. It exhibits osteoconductivity, which helps bone tissues to regenerate [3]. Additionally, it has the advantages of metals, including X-ray detection and ease of processing. Furthermore, its biodegradability eliminates the risk and cost of secondary surgery [4,5]. However, its poor mechanical properties, rapid degradation rate, and inflammatory effects on surrounding tissues due to hydrogen evolution limit its clinical applications [6,7].

According to previous reports, the mechanical properties of an Mg alloy can be enhanced by adding appropriate alloy elements [8], and the degradation rate can be slowed down via surface treatment [9]. In this study, the ZKX500 Mg–Zn–Zr–Ca alloy was selected. Zn can be added to improve the mechanical strength and corrosion resistance of Mg alloys [10], while Zr and Ca can be added to refine the grains [11,12,13,14]. Furthermore, the constant voltage anodic oxidation method was used to improve the biomedical compatibility of the material. It has been reported [15,16,17] that anodic oxidation film has a positive effect on the biomedical application of magnesium alloys. By producing a biocompatible magnesium oxide film, the degradation rate could be reduced, and thus, could achieve a better cell adhesion. However, there are few clinical studies about the biomedical application of anodic oxidation phosphate coating, which cannot effectively provide a reference for biomedical applications.

In the present work, an environmentally friendly anodic oxidation film on the ZKX500 Mg–Zn alloy was investigated. The tensile properties and impact toughness were investigated before and after immersion. Finally, biocompatibility testing and an animal experiment were conducted. From our pervious works [13], pure-Ti and Mg screws without surface treatment were compared in an animal experiment. Although the Mg screw presented better biocompatibility, the higher degradation rate could not afford the mechanical support. Thus, the introduction of phosphate coating is critical for a biodegradable material design. Relevant research results were used as a reference for the clinical application of magnesium matrix implants.

## 2. Materials and Methods

### 2.1. Materials and Specimen Preparation

A ZKX500 Mg–Zn–Zr–Ca alloy (Ting Sin Co., Ltd., Tainan, Taiwan) was used in this study. The as-received extruded materials are denoted as F. The anodic oxidation equipment was composed of a DC power supply (GW Instek ASR-2050R, Taiwan), a stainless-steel container, and a cooling system. Specimens (12 mm × 12 mm × 10 mm) were used as the anode, while the stainless-steel container was used as the cathode. Prior to the anodic oxidation treatment, the specimens were ground on SiC paper (P2500) and cleaned with acetone and alcohol. The phosphate-based electrolyte, which consisted of 8 g/L of sodium phosphate (Na_3_PO_4_·H_2_O), 1 g/L of sodium hydroxide (NaOH), and deionized water, was selected and the anodic oxidation was performed at a constant voltage of 250 V for 10 min. The anodic oxidation-treated specimens are denoted as FP.

### 2.2. Characterization of the Coatings

The microstructures of the coatings were observed by scanning electron microscopy (SEM) equipped with energy-disperse spectroscopy (EDS) (Hitachi SU-5000, Hitachi, Tokyo, Japan). The phase structures were an analysis by X-ray diffraction (XRD) (Bruker AXS GmbH, Karlsruhe, Germany) with CuK_α_ radiation operated at 45 kV. The scan rate of 3 °/min and 2θ range of 10°–90° were set in this study. Fourier transform infrared spectroscopy (FTIR) (PerkinElmer UATR Two, Waltham, Massachusetts, USA) was used to determine the formation of the coating layer. The coating surface was selected and cut with a focused ion beam (FIB) (FEI Nova 200, Hillsboro, Oregon, USA) for transmission electron microscopy (TEM) (JEOL JEM-2100F, Tokyo, Japan).

### 2.3. In Vitro Degradation Test and Mechanical Properties

The immersion test was conducted at 37 °C in simulated body fluid (SBF)—the composition is shown as Table 1 for 28 days according to the ASTM-G31 standard [18]. The corrosion rate can be calculated from the following equation:$$\mathrm{Corrosion}\;\mathrm{rate}\;(\mathrm{mm}/y)=87.6\times \left(W/DAT\right)$$
where W is the weight loss (mg), D is the density (1.74 g/cm^3^), A is the surface area (cm^2^), and the T is the immersion time (h). After the immersion test, the corrosion mechanism was evaluated by SEM and XRD. Tensile strength and impact toughness were measured after 14 and 28 days of immersion. The tensile test was performed on a universal testing machine (HT-8336, Hung Ta, Taichung, Taiwan) with a tensile rate of 1 mm/min. A Charpy impact testing machine (HUNG TA HT-8041A, Taichung, Taiwan) was used for the impact test. 

### 2.4. Biocompatibility Analysis

To investigate biocompatibility, a cell adhesion test was conducted using MG-63 cell lines (CRL-1427, ATCC, Manassas, Virginia, USA) according to the ISO 10993-5 standard [19]. Cells were seeded with a density of 8000 cells/well on FP specimen and placed in Dulbecco’s Modified Eagle Medium (DMEM) then incubated for 3 h and 24 h.

### 2.5. Implantation Test

Regarding the animal experiment, an anodic oxidation-treated ZKX500 cannulated bone screw was implanted in a Lanyu pig obtained from Taiwan. The animal experiment adhered to protocol approved by the Institutional Animal Care and Use Committee (IACUC) of the preclinical testing facility (NARLabs, Tainan, Taiwan). The implant area was the radiocarpal bone of the forelimb, and a fracture was made before implanted. CT scans were performed monthly for six months after surgery.

## 3. Results and Discussion

### 3.1. Micromorphology and Composition of the Coating

The surface morphologies and element compositions after anodic oxidation for 10 min at different voltages are shown in Figure 1. At 50 V and 150 V, the coating cracked, whereas at 250 V, a uniform surface was obtained. The surface morphologies and element compositions of anodic oxidation at 250 V for different treatment times are shown in Figure 2. The coating surface was melts with nano-pores, and the treatment duration did not affect the composition. The cross-sectional morphologies are shown in Figure 3. The density increased with the treatment time, and at about 10 min, the coating has the maximum thickness. The X-ray diffraction (XRD) patterns are shown in Figure 4. Additional peaks corresponding to the Mg_3_(PO_4_)_2_ and MgO phases were observed in the FP sample. Phosphate peaks also appeared in the Fourier-transform infrared (FTIR) spectra, ν_2_ O-P-O bending mode (567 cm^−1^) and antisymmetric P-O at 1039 cm^−1^ (Figure 5), confirming that the coating was made up of Mg_3_(PO_4_)_2_ and MgO [20,21,22].

The cross-sectional bright-field TEM image and EDS mapping results of the FP sample are shown in Figure 6. After anodic oxidation, it was observed that the sample could be divided into three layers (denoted as layers I, II, III). From EDS mapping, layer I was the phosphate coating that comprised O, Mg, and P; layers II and III were Mg substrate. As shown in Figure 7, ring diffraction patterns (DP) reveal that the coating mainly consists of amorphous magnesium phosphate and a small amount of polycrystalline magnesium [23,24]. HR-TEM shows an order region (yellow square region), which is confirmed to be plane {10$\bar{1}$0} through Fast Fourier Transformed (FFT) diffraction. Inverse FFT (IFFT) shows that the d-spacing of plane {10$\bar{1}$0} is about 2.79 Å. This also prove the existence of polycrystalline magnesium in the coating. From the DP, layers II and III were single crystalline magnesium (Figure 8), but layer II had less zinc concentration. Because of the more uniform phase distribution in the ion-exchanged layer than in the matrix, the coating and ion-exchanged layer are crucial for enhancing corrosion resistance [25,26,27]. Considering the low chemical activity of the coating, it can further protect the matrix from corrosion, and no potential difference in the uniform ion-exchanged layer can prevent galvanic corrosion.

### 3.2. Degradation Rates and Models

The results of immersion tests are shown in Figure 9. The low weight-loss percentage and degradation rate indicate the effectiveness of the anodic coating. XRD patterns and element compositions of samples after immersion are shown in Figure 10. The degradation products contain magnesium hydroxides and mainly consist of calcium and phosphorous oxides, which are similar to human bones, and thus, are beneficial for recovery (Figure 11). An overview of the degradation models for magnesium alloy and anodic coating in SBF based on the above results is shown in Figure 12.

### 3.3. Mechanical Properties after Immersion

Figure 13 presents the tensile curves measured after immersion in SBF, and the results of the strength, elongation, and toughness are shown in Figure 14 and Figure 15. After 14 days of immersion, the samples had reached the degradation product deposition stage, becoming more brittle, thereby increasing in strength while decreasing in ductility and toughness [28,29,30,31]. After 28 days of immersion, the coating had completely degraded, and the corrosion behaviors caused cracks that penetrated the core of the samples, thus, decreasing the tensile properties. However, the yield strength and ultimate tensile strength of the FP samples were higher than those of human bones. The tensile fracture surfaces are shown in Figure 16. The mechanism of the tensile fracture changed from ductile to brittle after 28 days of immersion, thus, decreasing the tensile properties. The impact toughness remained good after 28 days of immersion (Figure 17). More brittle characteristics on the fracture surfaces were observed after immersion (Figure 18), slightly decreasing the impact toughness [32,33].

### 3.4. Cell Adhesion

Figure 19a–d show the cell morphologies after 3 h of culturing. The cells on the F sample were spherical, while those on the FP sample were stretched and had an extension of lamellipodia. Figure 19e–h show the cell morphologies after 24 h incubation. The F sample had a relatively small adhesive area, whereas, for the FP sample, all the cells were attached to the surface. Hydrogen evolution through degradation and the roughness of the sample surface have been shown to affect the cell attachment [34,35]. Considering the SEM surface morphologies (Figure 1) and TEM BF cross-section (Figure 6), the anodic coating layer provided a rougher surface. Moreover, the lower corrosion rate and the hydrogen evolution led to better cell adhesion.

### 3.5. Animal Experiment

Figure 20 shows the surgical images and post-surgery CT scans. Figure 21 and Figure 22 show the follow-up from 1 to 6 months after surgery. Table 2 shows the remaining Mg screw length of each month. The screw thread could be observed after 1 month, and the black shade of the small area was due to hydrogen gas bubbles and image interference. After 3 months, the screw surface degraded, but the main structure of the screw remained intact. In general, the fracture recovery period was about 3 months. Compared to our previous work [13], in which the screw degraded by almost 40%, the anodic-coated magnesium bone screw was estimated to have appropriate support function and safety. After 4 to 6 months, the outline of the screw became blurry, indicating that the screw had integrated into the surrounding tissues. Furthermore, the experimental pig exhibited no adverse effects during recovery and could move with ease.

## 4. Conclusions

A uniform anodic coating mainly consisting of Mg_3_(PO_4_)_2_ with a thickness of about 800 nm was prepared by constant voltage anodic oxidation at 250 V for 10 min. After anodic oxidation, we have characterized the existence of the phosphate coating and an ion-exchange layer. The above observations serve as the main reason of the lower degradation rate and maintenance of mechanical properties.

In the tensile and impact tests, the coated magnesium alloy maintained high strength, ductility, and toughness after immersion. In the cell adhesion test and animal implantation experiment, the anodic coating also exhibited excellent biocompatibility.

## Figures and Tables

**Figure 1 bioengineering-09-00542-f001:**
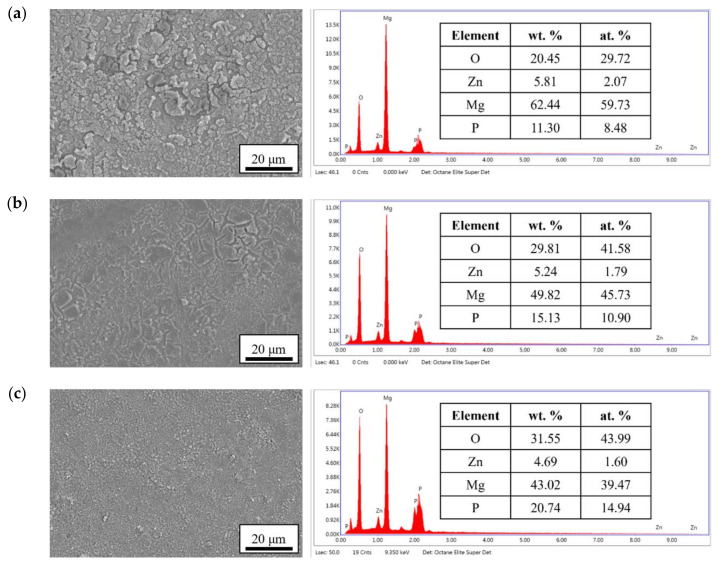
Surface morphologies and EDS spectra of the anodic coatings prepared at (**a**) 50 V, (**b**) 150 V, and (**c**) 250 V for 10 min.

**Figure 2 bioengineering-09-00542-f002:**
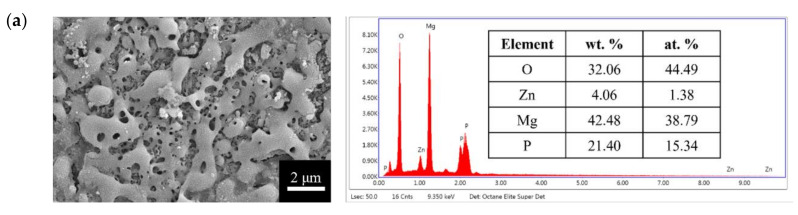
Surface morphologies and EDS spectra of the anodic coatings prepared at 250 V for (**a**) 5 min, (**b**) 10 min, and (**c**) 25 min.

**Figure 3 bioengineering-09-00542-f003:**
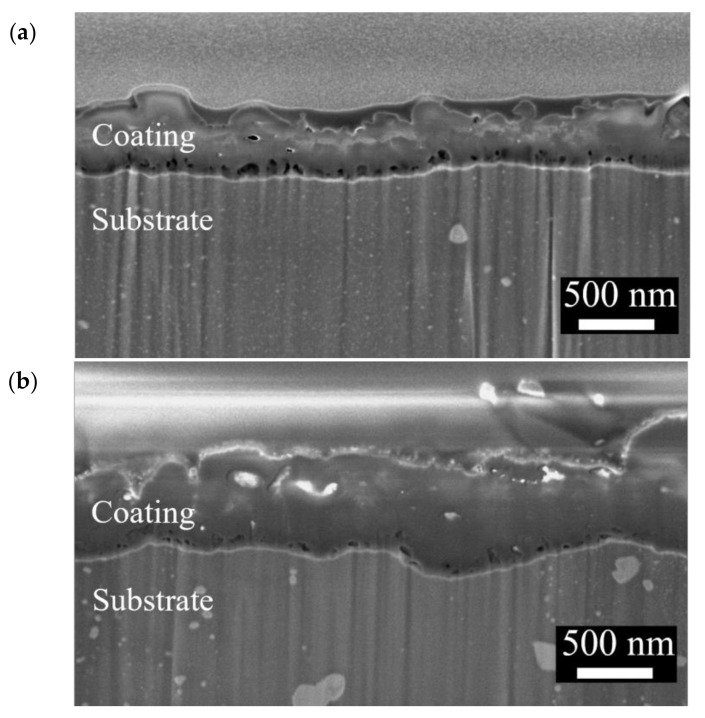
Cross-section morphologies of the anodic coatings prepared at 250 V for (**a**) 5 min, (**b**) 10 min, and (**c**) 25 min.

**Figure 4 bioengineering-09-00542-f004:**
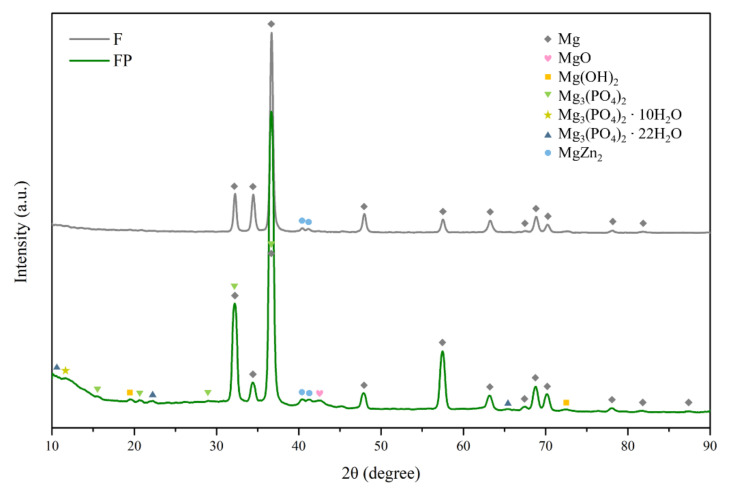
XRD patterns of F and FP.

**Figure 5 bioengineering-09-00542-f005:**
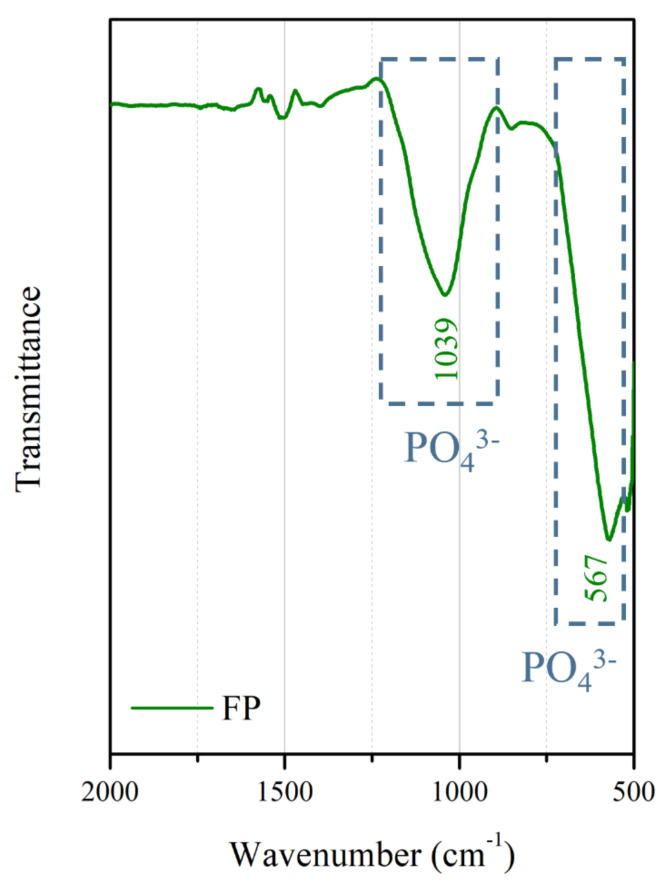
FTIR spectrum of the anodic coating.

**Figure 6 bioengineering-09-00542-f006:**
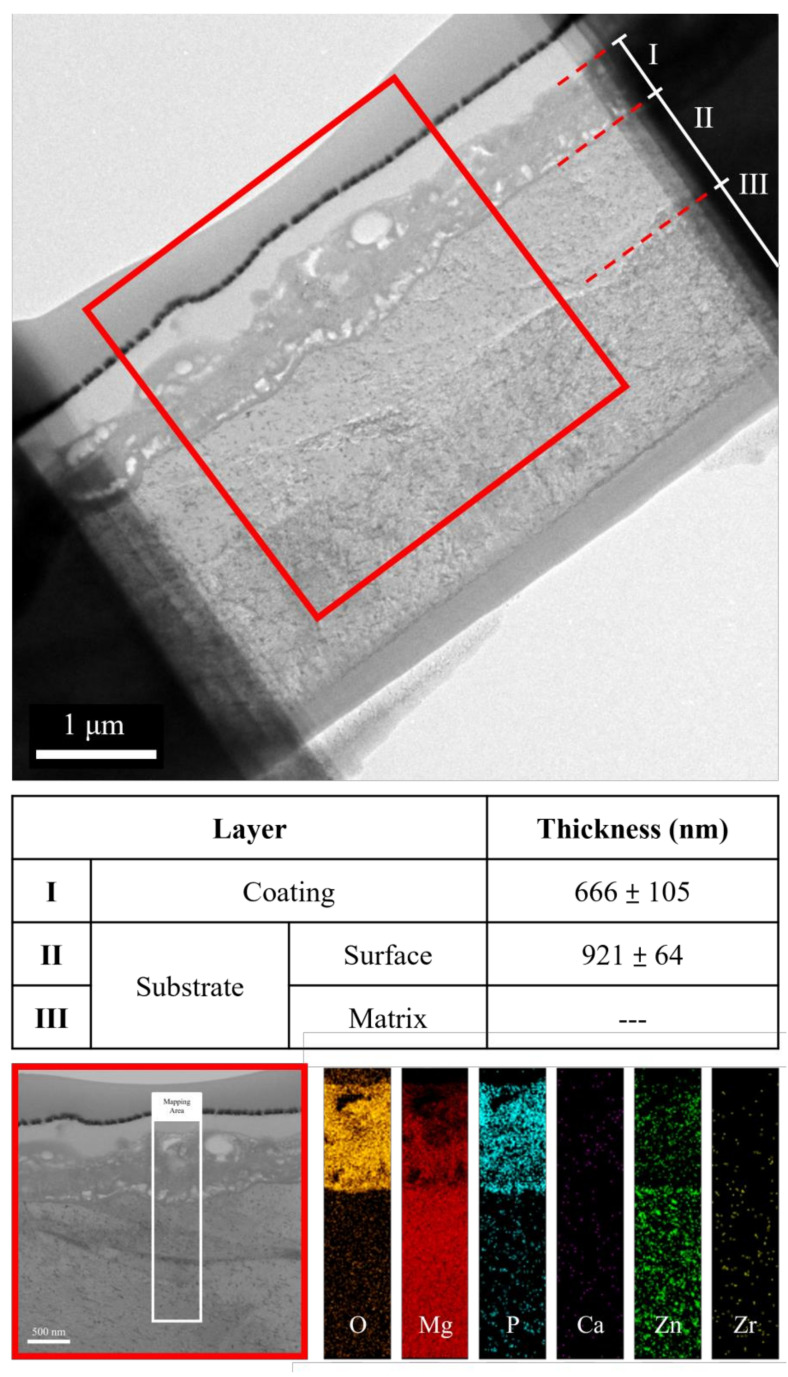
TEM image and EDS mapping of the anodic coating cross-section.

**Figure 7 bioengineering-09-00542-f007:**
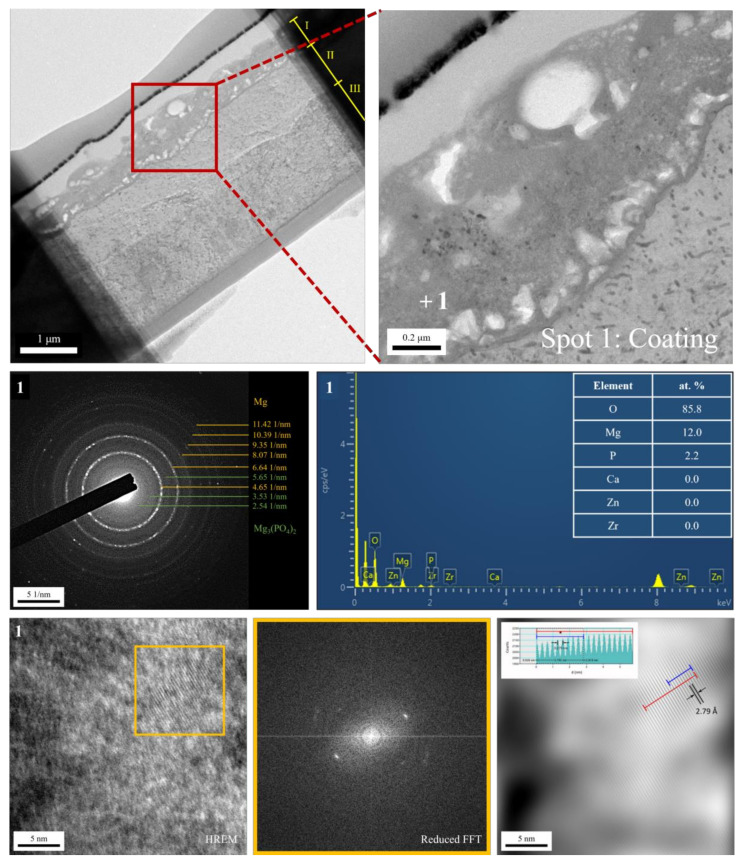
TEM analysis of the anodic coating.

**Figure 8 bioengineering-09-00542-f008:**
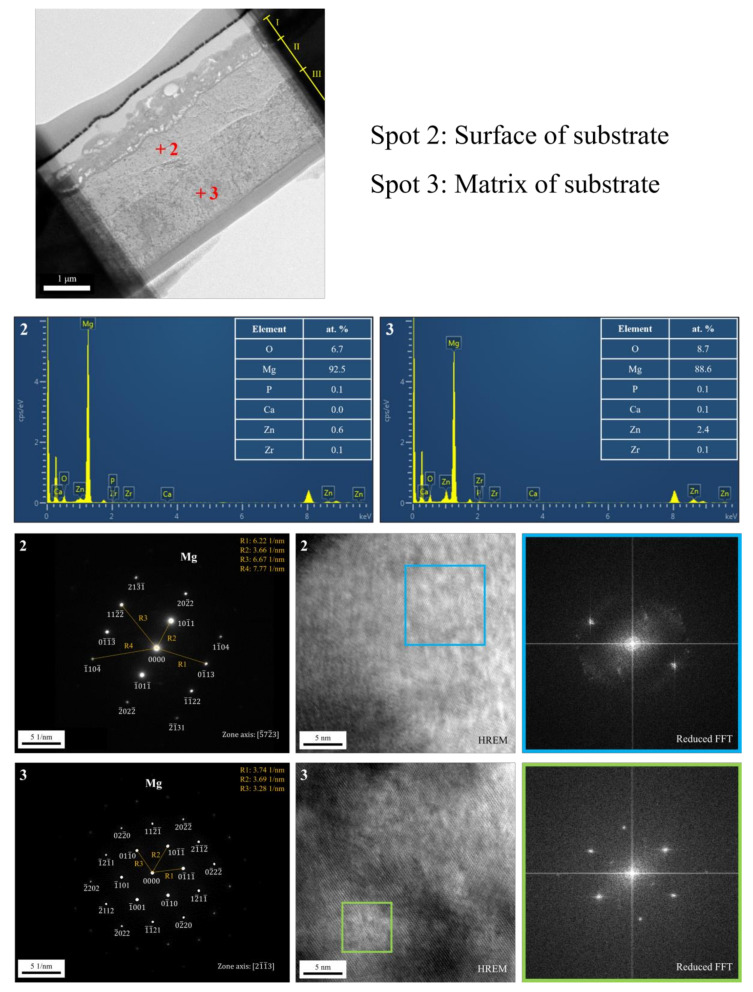
TEM analysis of the substrate after anodic oxidation.

**Figure 9 bioengineering-09-00542-f009:**
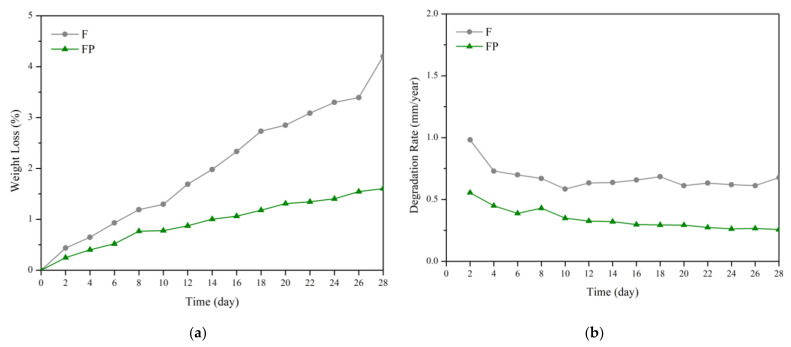
(**a**) Weight loss percentages and (**b**) degradation rates of F and FP in the SBF immersion test.

**Figure 10 bioengineering-09-00542-f010:**
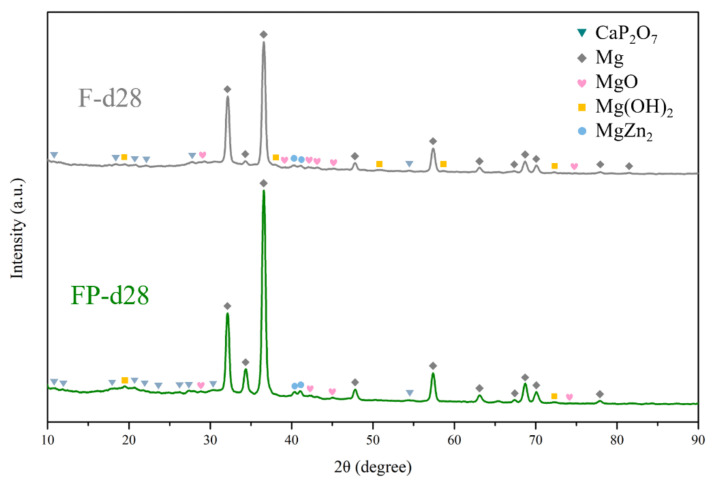
XRD patterns of F and FP after immersed in SBF for 28 days.

**Figure 11 bioengineering-09-00542-f011:**
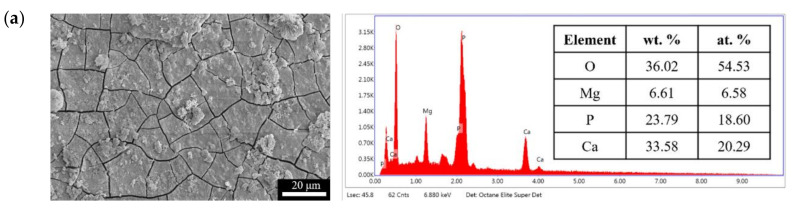
Surface morphologies and EDS spectra of (**a**) F and (**b**) FP after immersed in SBF for 28 days.

**Figure 12 bioengineering-09-00542-f012:**
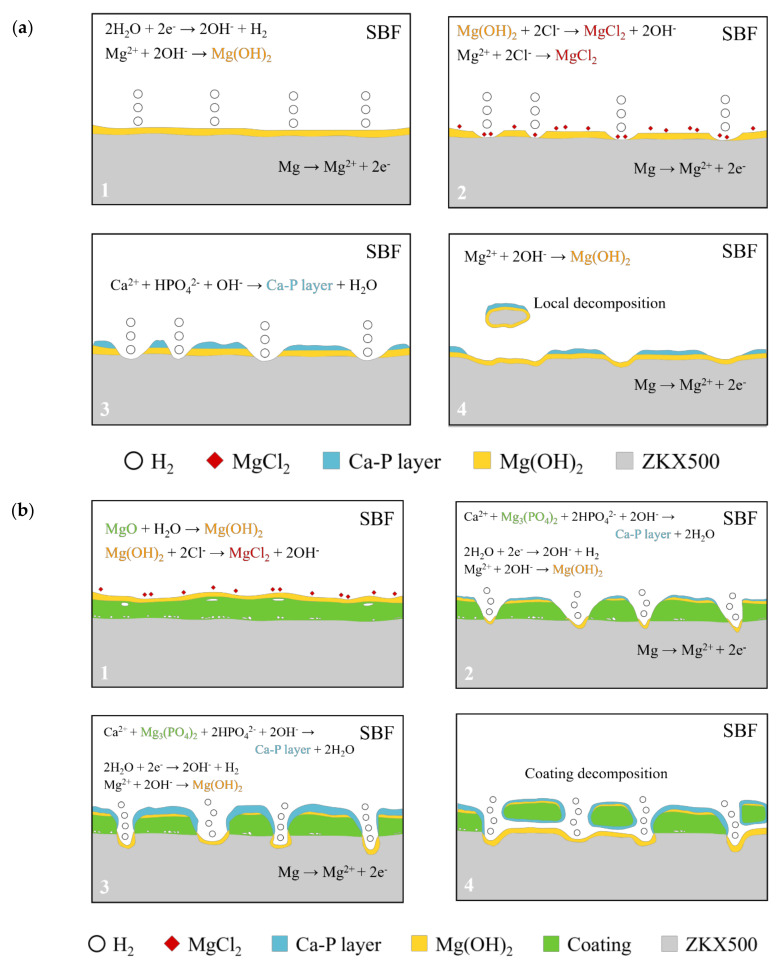
Degradation mechanism of (**a**) F and (**b**) FP in SBF.

**Figure 13 bioengineering-09-00542-f013:**
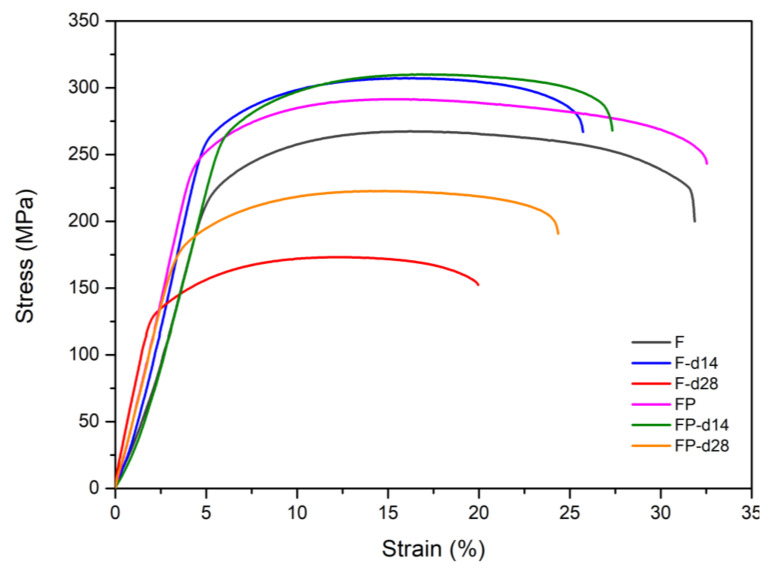
Stress–strain curves of F and FP before and after immersed in SBF.

**Figure 14 bioengineering-09-00542-f014:**
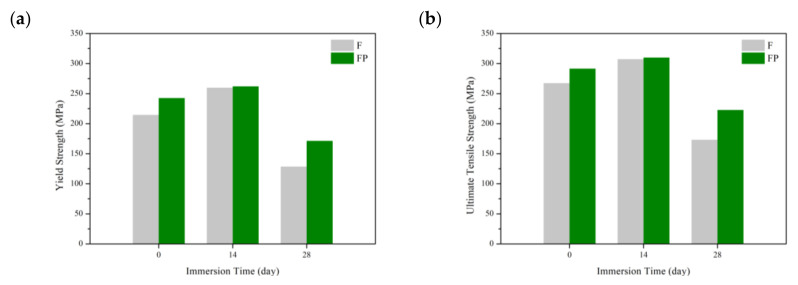
Tensile properties of F and FP before and after immersed in SBF: (**a**) yield strength, (**b**) ultimate tensile strength, (**c**) uniform elongation, and (**d**) tensile elongation.

**Figure 15 bioengineering-09-00542-f015:**
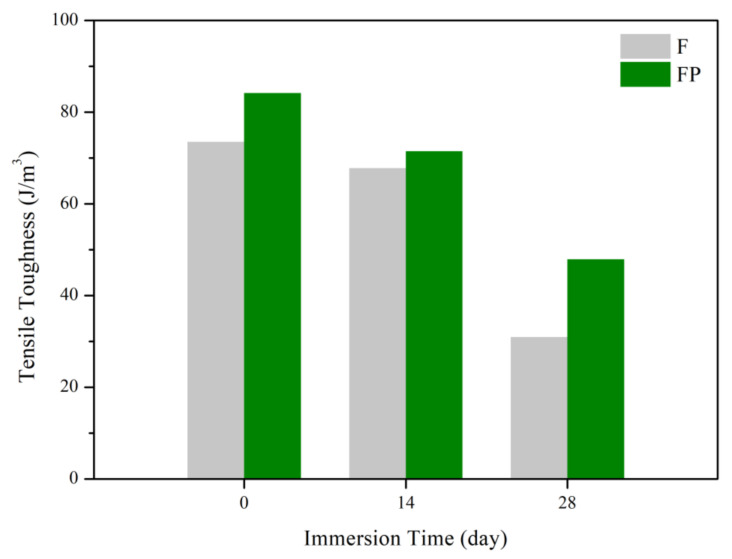
Tensile toughness of F and FP before and after immersed in SBF.

**Figure 16 bioengineering-09-00542-f016:**
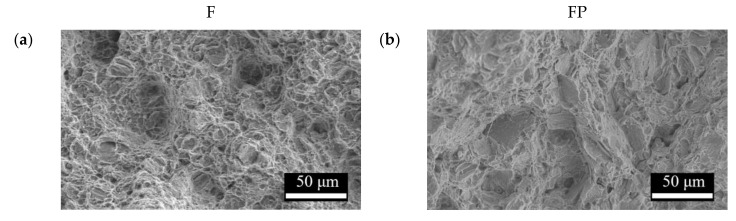
Tensile fracture surface morphologies of (**a**) F, (**b**) FP, (**c**) F-d28, and (**d**) FP-d28.

**Figure 17 bioengineering-09-00542-f017:**
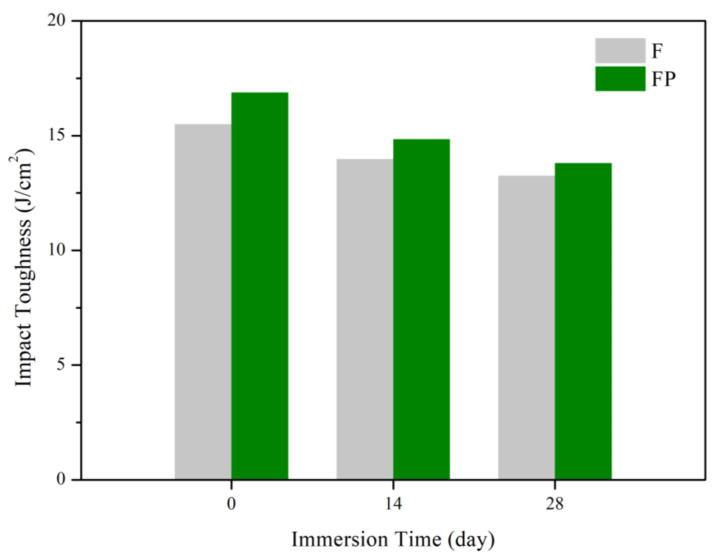
Impact toughness of F and FP before and after immersed in SBF.

**Figure 18 bioengineering-09-00542-f018:**
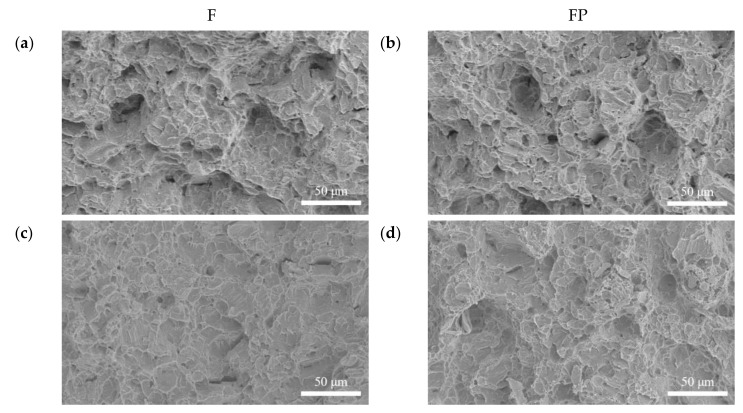
Impact fracture surface morphologies of (**a**) F, (**b**) FP, (**c**) F-d28, and (**d**) FP-d28.

**Figure 19 bioengineering-09-00542-f019:**
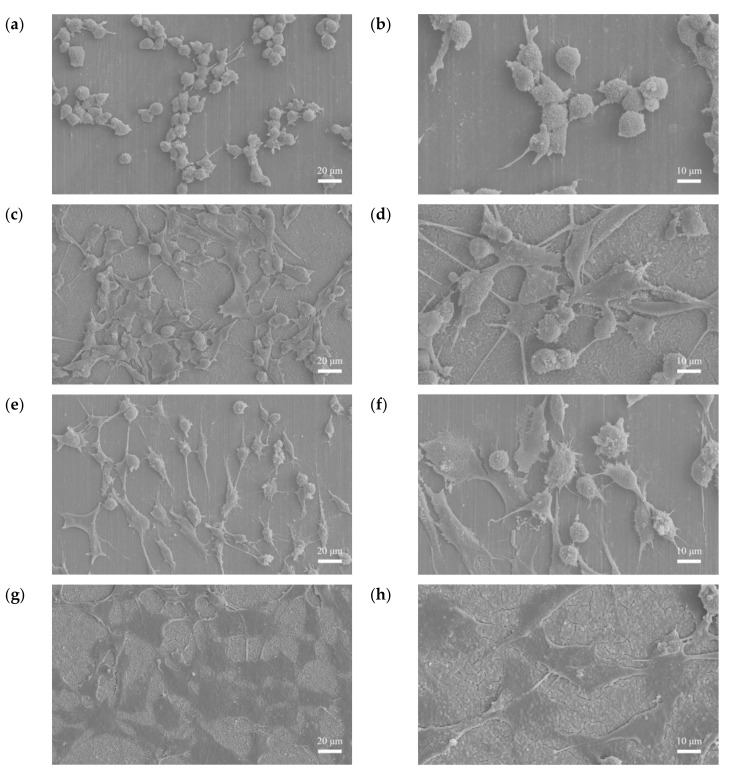
Cell adhesion results: (**a**,**b**) F-3 h, (**c**,**d**) FP-3 h, (**e**,**f**) F-24 h, and (**g**,**h**) FP-24 h.

**Figure 20 bioengineering-09-00542-f020:**
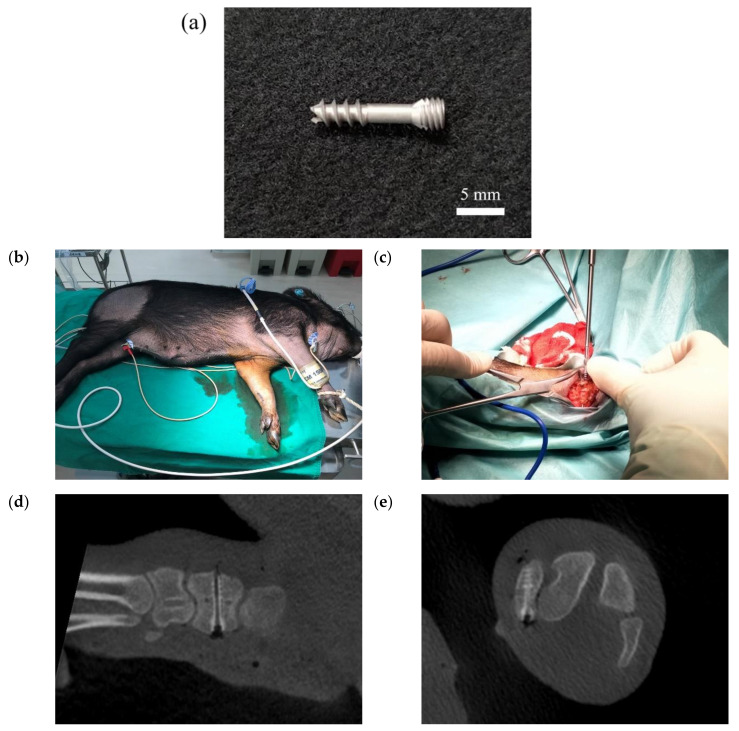
The (**a**) magnesium bone screw, (**b**) experimental animal, (**c**) implantation process, and (**d**,**e**) post-surgery CT scans in the animal experiment.

**Figure 21 bioengineering-09-00542-f021:**
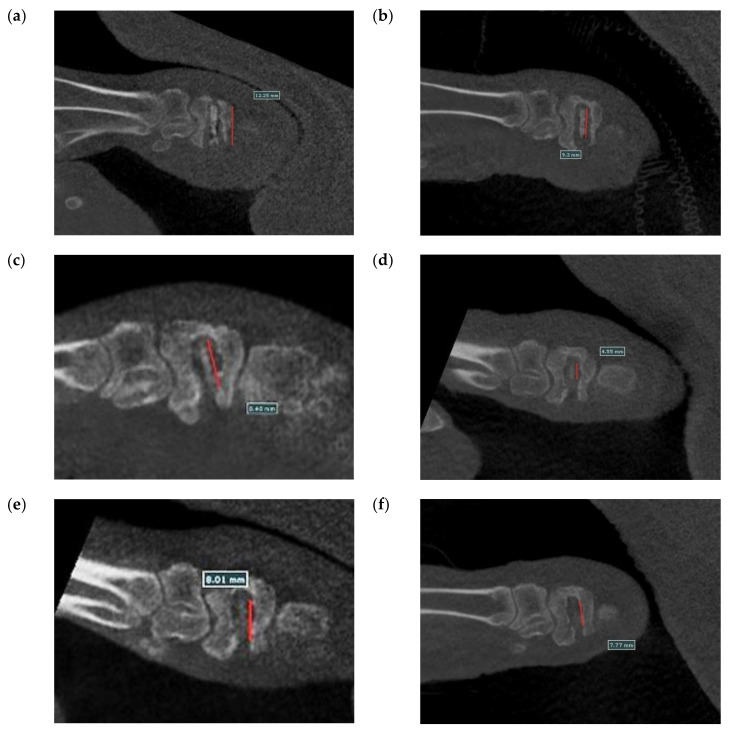
Longitudinal section CT scans recorded (**a**) 1 month, (**b**) 2 months, (**c**) 3 months, (**d**) 4 months, (**e**) 5 months, and (**f**) 6 months after surgery.

**Figure 22 bioengineering-09-00542-f022:**
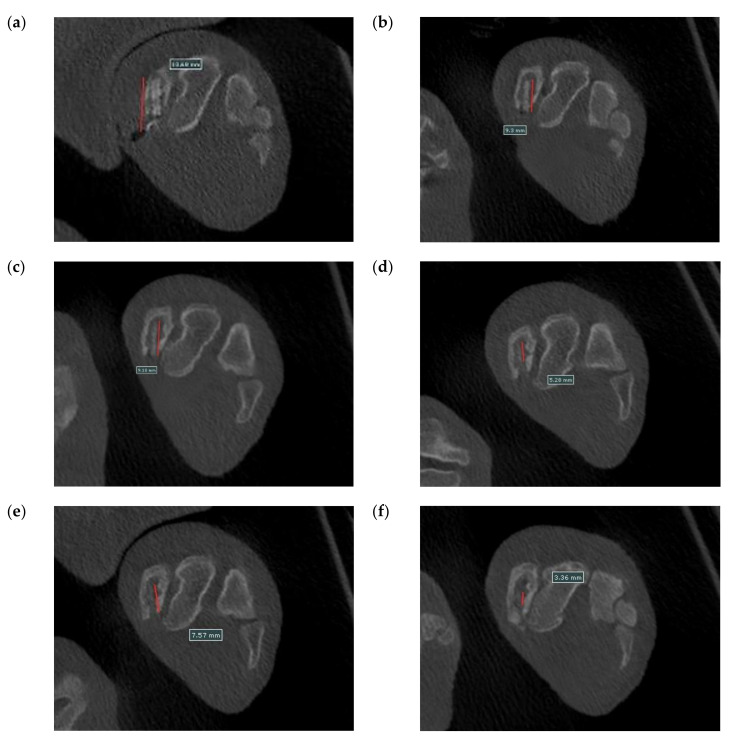
Cross-section CT scans recorded (**a**) 1 month, (**b**) 2 months, (**c**) 3 months, (**d**) 4 months, (**e**) 5 months, and (**f**) 6 months after surgery.

**Table 1 bioengineering-09-00542-t001:** Composition of the SBF.

Compound	Amount (g/L)
NaCl	5.403
NaHCO_3_	0.740
Na_2_CO_3_	2.046
KCl	0.225
K_2_HPO_4_	0.230
MgCl_2_·6H_2_O	0.311
HEPES	11.928
CaCl_2_	0.294
Na_2_SO_4_	0.072
NaOH	Buffer to pH 7.4

**Table 2 bioengineering-09-00542-t002:** The remain length of the Mg screw.

Months	Mg Screw Length (mm)
1	12.25
2	9.30
3	9.18
4	8.98
5	8.01
6	7.77

## Data Availability

The data presented in this study are available on request from the corresponding author. The data are not publicly available due to privacy or ethics.

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
