# Peer review of "A Study on the Structure and Biomedical Application Characteristics of Phosphate Coatings on ZKX500 Magnesium Alloys"

_bioengineering, 2022, doi:10.3390/bioengineering9100542_

Round 1
Reviewer 1 Report
1. Abstract. “Additionally, its bioidegradability…” should be “Additionally, Mg biodegradability…”
2. Abstract. “animalimplantation” should be “animal implantation”
3. Keywords. Replace “degradation” by “corrosion”. Also, consider adding “implant” as a keyword.
4. Introduction. “…than those of other metals…” should be “than other metals”.
5. The introduction is too short. There are many studies about the biocompatibility of Mg-Zn alloys with anodic films. The authors should provide a brief background on the subject and highlight the novelty of their study.
6. Materials and methods. “…were polished…” should be “…were ground…”. Please indicate the number of the final SiC paper (P1200?).
7. Materials and methods. The electrolyte for anodic oxidation should be specified. The size of the specimens and current density limit should also be indicated. The volume of electrolyte, counterelectrode and power supply should also be mentioned. SBF: The authors should describe acronyms when they are first used in the manuscript.
8. Where the specimens carbon coated or gold coated for SEM examination? If gold-coated, then EDS spectra should be revised. Gold and phosphorous peaks are in the same region.
9. Results. “…to have melts…” This is poor scientific writing.
10. Results. FTIR peaks should be better described. Bending, stretching, rocking of P-O bonds?
11. Results. TEM specimen preparation should be described in the Materials and Methods section.
12. Results. “After anodic oxidation, 3 layers were formed on the sample surface, including coating, surface, and matrix” This sentence does not make sense. Please rewrite.
13. Results. “The coating comprised O, Mg, and P.” What about Zn?
14. Results. “As shown in Figure 7, the coating mainly consists of amorphous magnesium phosphate and a small amount of polycrystalline magnesium”. This is inconsistent with XRD results. The presence of metallic magnesium in the anodic film is doubtful. MgO is more likely.
15. Results “ion-exchanged layer” This term is confusing. What do the authors mean by ion-exchanged layer?
16. Results. “…galvanic corrosion was prevented…” This sentence needs further explanation. Galvanic corrosion involves anodes and cathodes. Please elaborate further.
17. Figure 4. XRD should be performed at a glancing angle. Peaks assignment is not convincing (small intensity peaks for all the coating’s constituents).
18. XRD and FTIR should be briefly described in the Materials and Methods section.
19. Figure 6. The layer designated as II seems to be an artifact from specimen preparation. I am not convinced that this is an additional layer.
20. The description of Figure 7 is incomplete. For instance, what do the authors get from the 2.79 value in the last figure.
21. Figure 8. The difference between Zn content between 2 and 3 should be confirmed by other means. For instance, the authors should observe a similar result by performing EDS analyses on the SEM cross-sections.
22. Figure 9. Weight loss measurements should be better described in the Materials and Methods section. How did the authors remove the corrosion products? How did the authors measure the weight loss of FP specimens without damaging the coating?
23. Figure 10. I don’t believe the XRD results. Many of the peaks are almost invisible. Glancing angle should be used to increase the signal for surface layers.
24. Figure 11b. It seems that the anodic film is gone. The authors should provide SEM cross-sections after 28 days. Otherwise, there is no evidence to support Figure 12.
25. Results. “corrosion behaviors had penetrated inside the samples” this sentence does not make sense. It should be rewritten.
26. Figure 13. The choice of colors could be improved. It is difficult to distinguish the curves.
27. Figure 14. A scatter plot is recommended.
28. Figure 16. What causes the transition from ductile to brittle fracture?
29. Figure 17. What is the difference between figures d and e?
30. Conclusions. The thickness of the coating is only mentioned in the conclusions. This information should be included in the body of the text.
Author Response
See attached.
Response to Reviewer 1 Comments
- Abstract. “Additionally, its bioidegradability…” should be “Additionally, Mg biodegradability…”
We have corrected. - Abstract. “animalimplantation” should be “animal implantation”
We have corrected text errors. - Keywords. Replace “degradation” by “corrosion”. Also, consider adding “implant” as a keyword.
We have added implant as a keyword, however, corrosion is used for industry and degradation is used for medicine, we will remain the keyword. - Introduction. “…than those of other metals…” should be “than other metals”.
We have corrected text errors. - The introduction is too short. There are many studies about the biocompatibility of Mg-Zn alloys with anodic films. The authors should provide a brief background on the subject and highlight the novelty of their study.
We have added explanations. - Materials and methods. “…were polished…” should be “…were ground…”. Please indicate the number of the final SiC paper (P1200?).
We have added explanations. - Materials and methods. The electrolyte for anodic oxidation should be specified. The size of the specimens and current density limit should also be indicated. The volume of electrolyte, counterelectrode and power supply should also be mentioned. SBF: The authors should describe acronyms when they are first used in the manuscript.
We have added explanations. - Where the specimens carbon coated or gold coated for SEM examination? If gold-coated, then EDS spectra should be revised. Gold and phosphorous peaks are in the same region.
The specimens were carbon coated. - Results. “…to have melts…” This is poor scientific writing.
We have improved the writing. - Results. FTIR peaks should be better described. Bending, stretching, rocking of P-O bonds?
FTIR peaks were bending of P-O bonds, we have added the description in revised. - Results. TEM specimen preparation should be described in the Materials and Methods section.
We have added explanations in revised. - Results. “After anodic oxidation, 3 layers were formed on the sample surface, including coating, surface, and matrix” This sentence does not make sense. Please rewrite.
3 layeas were detected by TEM and we have revised. - Results. “The coating comprised O, Mg, and P.” What about Zn?
From our experiment and others reference, under high voltage, magnesium transforms into ion and migrate outward to the surface and form the compound. So, this is the difference in bonding energy. - Results. “As shown in Figure 7, the coating mainly consists of amorphous magnesium phosphate and a small amount of polycrystalline magnesium”. This is inconsistent with XRD results. The presence of metallic magnesium in the anodic film is doubtful. MgO is more likely.
From the diffraction pattern, we calculate the d-spacing and it matches better with magnesium than MgO, and the XRD show small intensive of MgO peak, this is the main reason the magnesium oxide is thinner. - Results “ion-exchanged layer” This term is confusing. What do the authors mean by ion-exchanged layer?
Under high voltage, magnesium will transform into ion and migrate, so as oxygen and phosphate. The chemical reaction happens at the surface and lead to a layer which composition is slightly different to matrix. So, this layer is called: ion-exchanged layer. - Results. “…galvanic corrosion was prevented…” This sentence needs further explanation. Galvanic corrosion involves anodes and cathodes. Please elaborate further.
We have added explanations in revised. - Figure 4. XRD should be performed at a glancing angle. Peaks assignment is not convincing (small intensity peaks for all the coating’s constituents).
The coating layer is very thin, although XRD was performed at a glancing angle, the intensity is still low. So, this study used TEM for further investigation. - XRD and FTIR should be briefly described in the Materials and Methods section.
We have added explanations in revised. - Figure 6. The layer designated as II seems to be an artifact from specimen preparation. I am not convinced that this is an additional layer.
TEM specimen was prepared by FIB, and it won’t cause any artifact. The layer II was formed during the anodic oxidation. - The description of Figure 7 is incomplete. For instance, what do the authors get from the 2.79 value in the last figure.
We have revised. - Figure 8. The difference between Zn content between 2 and 3 should be confirmed by other means. For instance, the authors should observe a similar result by performing EDS analyses on the SEM cross-sections.
Figure 8 has shown the STEM-EDS analysis and both diffraction pattern. STEM-EDS has rather higher atomic concentration resolution than normal SEM-EDS because of electron bean probe-size, we applied TEM for further investigation of precise Zn atomic-ratio. These data are precise and error-free. - Figure 9. Weight loss measurements should be better described in the Materials and Methods section. How did the authors remove the corrosion products? How did the authors measure the weight loss of FP specimens without damaging the coating?
We have revised. - Figure 10. I don’t believe the XRD results. Many of the peaks are almost invisible. Glancing angle should be used to increase the signal for surface layers.
This XRD was performed after immersion for 28 days, and the coating layer is almost gone. Glancing angle had been used but still the signal is low. (effects of surface compounds) - Figure 11b. It seems that the anodic film is gone. The authors should provide SEM cross-sections after 28 days. Otherwise, there is no evidence to support Figure 12.
Our group have investigated Mg alloy for a long time, no matter in engineering or in biomaterial, based on our previous reports and other literature, we can conclude a general corrosion mechanism for Mg alloy.
- J. Lin, F. Y. Hung*, M. L. Yeh, H. P. Lee, T. S. Lui, Correlation Between Anti-Corrosion Performance and Optical Reflectance of Nano Fluoride Film on Biodegradable Mg-Zn-Zr Alloy: A Non-Destructive Evaluation Approach, International Journal of Electrochemical Science, 12 (2017) 3614-3634 (ELECTROCHEMISTRY, 19/29=65.5%, IF=1.469)
- J. Lin, F. Y. Hung*, M. L. Yeh, H. P. Lee, T. S. lui, Development of A Novel Micro-Textured Surface Using Duplex Surface Modification for Biomedical Mg Alloy Applications, Materials Letters, 206 (2017) 9-12. (MATERIALS SCIENCE, MULTIDISCIPLINARY, 90/275=32.7%, IF=2.572)
- J. Lin, F. Y. Hung*, H. J. Liu, M. L. Yeh, Dynamic Corrosion and Material Characteristics of Mg-Zn-Zr Mini-tubes: The Influence of Microstructures and Extrusion Parameters, Advanced Engineering Materials, 19(11) (2017) 1700159. (MATERIALS SCIENCE, MULTIDISCIPLINARY, 103/275=37.4%, IF=2.319)
- J. Lin, F. Y. Hung*, M. L. Yeh, Development of a Novel Degradation-Controlled Magnesium-Based Regeneration Membrane for Future Guided Bone Regeneration (GBR) Therapy, Metals, 7(11) (2017) 481. (METALLURGY & METALLURGICAL ENGINEERING, 13/74=17.5%, IF=1.984)
- T. Chen, F. Y. Hung*, J. C. Syu, Biodegradable Implantation Material: Mechanical Properties and Surface Corrosion Mechanism of Mg-1Ca-0.5Zr alloy, Metals, 9(8) (2019) 857. (METALLURGY & METALLURGICAL ENGINEERING, 18/76=23.6%, IF= 2.259)
- T. Chen, F. Y. Hung*, T. S. Lui, Y. L. Lin, C. Y. Lin, Biodegradation ZK50 Magnesium Alloy Compression Screws: Mechanical Properties, Biodegradable Characteristics and Implant Test, Journal of Orthopaedic Science, 25(6) (2020) 1107-1115. (ORTHOPEDICS, 60/82=73%, IF= 1.259)
- T. Huang, F. Y. Hung*, F. C. Kuan, K. L. Hsu, W. R. Su, C. Y. Lin, Microstructure, Mechanical Properties, Degradation Behavior, and Implant Testing of Hot-Rolled Biodegradable ZKX500 Magnesium Alloy, Applied Sciences, 11(22) (2021) 10677. (ENGINEERING, MULTIDISCIPLINARY, 92/170=54%, IF=2.679)
- J. Chen, F. Y. Hung*, Y. T. Wang, C. W. Yen, Mechanical Properties and Biomedical Application Characteristics of Degradable Polylactic Acid-Mg-Ca3(PO4)2 Three-Phase Composite, Journal of the Mechanical Behavior of Biomedical Materials, 125 (2022) 104949. (ENGINEERING, BIOMEDICAL, 48/98=48%, IF=4.042)
25. Results. “corrosion behaviors had penetrated inside the samples” this sentence does not make sense. It should be rewritten.
The sentence had been rewritten.
26. Figure 13. The choice of colors could be improved. It is difficult to distinguish the curves.
We have changed the colors of figure 13 in revised paper.
27. Figure 14. A scatter plot is recommended.
Figure 14 contain 4 tensile properties; we think it’s better to plot in bar.
28. Figure 16. What causes the transition from ductile to brittle fracture?
The formation of microcracks during corrosion at as the fracture origin point cause the crack to propagate more easily.
29. Figure 17. What is the difference between figures d and e?
There are no figures d and e in figure 17. In figure 18, d and e were in different magnification and also different matrix, they are all illustrate in figure legend.
30. Conclusions. The thickness of the coating is only mentioned in the conclusions. This information should be included in the body of the text.
The thickness was mentioned in figure 6, explanation will add and emphasize in revised paper.

Reviewer 2 Report
See Attached

Author Response
see attached.
Response to Reviewer 2 Comments
1. Methods are insufficiently described:
a) Need commercial (or other) source of alloy.
b) Need composition of the phosphate-based electrolyte.
c) Need composition of SBF.
d) Need details on how the MG-63 cell line was cultured: source of cell line, media, incubator, etc.
e) Need better description of the source of the pig. Need to state whether the methods were done after approval of an institutional Animal Care and Use Committee or equivalent.
f) Need a description of the animal surgery. At the very least name the bone that the screw was inserted across.
g) Need descriptions of all of the machines used in analysis. TEM, SEM with EDS, XRD, FTIR machine. Type, manufacturer, state/province, country.
h) Need a description of preparation of samples for analysis, cell culture and immersion.
i) How were the samples were immersed in SBF? Were the samples suspended, or was only the top surface exposed to SBF?
We have added description of method in revised.
2. Did the authors use the screw that was used for in vivo use for any of the metal characterization studies? Just producing a screw can drastically change all of the metal properties of Mg. So, it would have been good if at least some of the characterization was checked between a solid piece of Mg and the screw. Did the authors at least do an immersion test comparing a screw without anodization to one that was anodized?
In our previous work we have investigated the properties of the screw, in this work, we focus on the effects of the coating layer.
3. In figures 1-3, 6, 7, 8, 11, 16, 18, 19, the images appear to be taken with a scanning electron microscope (SEM) and EDS is done with in the same way as SEM: bouncing electrons or X-rays off the surface of the specimen. With a transmission electron microscope (TEM), which is what the authors’ state they use, the specimen must be thin enough for electrons to penetrate. If this is TEM, how did the authors prepare their specimens to get them thin enough to do TEM? Please clarify.
The specimen for TEM is prepared by FIB, we have added explanations in revised.
4. Page 2, last sentence on the page, line 80: The sentence suggests that the authors are saying that they found that their coating prevented galvanic corrosion. They did not prove this. The sentence needs to be re-worded such that it is clear that they are talking about what other authors and studies have found, and that perhaps they are speculating that their coating similarly may have reduced chemical activity and galvanic corrosion.
We had revised.
5. The same criticism of the sentence on page 15, section 3.4, starts on line 148. Are the authors talking about their findings or findings from the literature? Clarify.
It’s our findings similar to others literature. We have rewritten the sentence and added explanations in revised.
6. Define what the authors mean by the “fracture recovery period”. How did they define fracture recovery?
Before the implant the osteotomy was performed.
we have added description in revised.
7. What are the very tiny numbers in the CT images? The authors never talk about them. They are extremely hard to read. What story do they tell?
The tiny numbers were just the measurement about the length of Mg screw, they help to know how much Mg screw had remain.
8. As there were absolutely no controls for the animal experiment, the authors have to be very cautious about making conclusions in terms of function. They state that the screw had “appropriate support function”. What do they mean? This screw was not placed in a spot where it was weight bearing, so it was not supporting weight. They don’t have a control that shows whether the “fracture recovery period” was better or worse than it would be if a screw had been inserted and then removed. Or if a Ti screw had been used. They also did not compare this to insertion of the uncoated screw, so there is no comparison to determine if the anodization had any effect over a non-anodized screw in terms of how long it took to degrade in vivo.
We have investigated the difference between Ti screw and the uncoated screw in our previous works.
9. Define biocompatibility: Although their pictures show no gross abnormalities, they should describe in words, how their images show biocompatibility. Consult a veterinarian for good terms. You have to define it in terms of the CT, because the authors did not do any histology. If a specialist in CT can describe the recovery, then that would be better.
This article was focus on the effect of phosphate coating. After discussed with the doctors, only CT data show in the article. We will add histology in our further study.
10. References: There is a concern that the extensive body of literature arising in Europe on Mg and Mg degradation has not been cited. While there is excellent metallurgy done in Asia, there is equally excellent work done in other areas of the world. I find the literature sources a bit limited. Also, except for a few reviews from the last three years, most of the references are moderately older.
We had added some references in revised paper.

Reviewer 3 Report
The manuscript bioengineering-1888743 was reviewed. The authors need to address the following comment before the publication:
Did authors consider any porous Mg samples for this application? In either case, It is better to provide some literature review on the Mg foams with the similar bio application in the manuscript.
Author Response
see attached.
Did authors consider any porous Mg samples for this application? In either case, It is better to provide some literature review on the Mg foams with the similar bio application in the manuscript.
We have search for literature of Mg foams, of course it has excellent properties, but the bone implants require higher mechanical properties and the processing applicability of Mg foams is not ideal. We think porous Mg is not a proper choice for bond screw. Notably, neural scaffolds and dental regeneration membranes are suitable.

Round 2
Reviewer 1 Report
1. “Tunneling scanning microscope” should be “transmission electron microscope”.
2. Figure 7. The presence of Mg in the PEO coating is very doubtful. Its presence in the SAED pattern may be explained by a relatively large analyzed area that includes some portion of the substrate. Please correct or show the size of the SAED analysis.
3. Results. “…due to the uniform phase distribution in the ion-exchanged layer, which was more than that in the matrix”. This sentence needs rewriting. The message is not clear.
4. “…no potential different in the uniform ion-exhanged layer, galvanic corrosion was prevented”. This sentence has spelling errors.
5. Answer to comment 22 is not satisfactory. Answer to comment 28 is not satisfactory.
Author Response
Please see attached.
Response to Reviewer
Reviewer 1
- “Tunneling scanning microscope” should be “transmission electron microscope”.
We have corrected.
- Figure 7. The presence of Mg in the PEO coating is very doubtful. Its presence in the SAED pattern may be explained by a relatively large analyzed area that includes some portion of the substrate. Please correct or show the size of the SAED analysis.
The probe size of the TEM is about 0.5 nm, and the coating is about 1 μm, we don’t think the analyzed area would include substrate.
- “…due to the uniform phase distribution in the ion-exchanged layer, which was more than that in the matrix”. This sentence needs rewriting. The message is not clear.
We had rewritten the sentence.
- “…no potential different in the uniform ion-exhanged layer, galvanic corrosion was prevented”. This sentence has spelling errors.
We had corrected.
- Answer to comment 22 is not satisfactory. Answer to comment 28 is not satisfactory.
Comment 22: We had added the method of the immersion test.
Comment 28: From the figure 14, it showed that the elongation was decrease (metal materials: decreasing ductility equals increasing brittleness) and the morphologies also show the characteristic similar to brittle fracture, so we propose that the fracture mechanism has changed.
.

Reviewer 2 Report
In this revision, the authors have made some major improvements. However, a few points are still concerning.
1. The authors did add more adequate descriptions of the methods, including institutional approval of animal use. Thank you.
2. The authors changed added the definition of TEM to be tunneling EM in the methods. But in the Results, section 3.1, page 3, line 117, they redefine TEM to be transmission EM. I believe that this might be a mistake.
3. Results, section 3.1, page 3, last sentence, lines 131-132. The last sentence does not make sense in English. Plus they give references in a sentence that appears to say that they have proven something. Suggestion: re-write the last 3 sentences and put the references in the right place and clarify the rationale. This is important because they want to say this in the conclusion.
4. Is it ion-exchanged or ion-exchange layer? Check and use only one version throughout the paper. (My preference would be ion-exchange layer, but there may be a convention in engineering.)
5. Figure 12. The diagrams suggest to me that there is more Mg degradation with the coating than without because the corrosion pits are more extensive. Is this what the authors want to imply?
6. Results, section 3.4, page 15, lines 198-199. The authors say: “Hydrogen evolution through degradation and the roughness of the sample surface affected the cell attachment [34-35].” This sounds like they are talking about their own findings, but then they give references. I believe that the authors are trying to say this? Hydrogen evolution. …sample surface has been shown to affect cell attachment [34-35].
7. The next (last) sentence would benefit from making it clear that the authors are speculating. For example: We suggest/speculate/propose that the anodic coating improved cell attachment by improving corrosion resistance, thereby reducing hydrogen release, and increasing surface roughness compared to other metals.
8. Response to comment 7: The small numbers in the CT images should be made into a table to support their comments about the changes in Mg degradation over time. They appear to have used the data to make statements, so please show the data in a form that can be read more easily.
9. The authors state that they have looked at Ti and uncoated Mg screw implantation previously. This paper would benefit from a discussion in which they compare this pig experiment to their previous work, in terms of function or changes in screw degradation.
10. Conclusion: The authors say: “the phosphate coating and ion-exchange layer were found to be the keys to slowing down the degradation and maintaining the mechanical properties.” You can’t say that these were keys because that suggests that you “proved” this. Aren’t the authors speculating? A suggested alternative statement might be: After anodic oxidation, we have characterized the existence of the phosphate coating and an ion-exchange layer, which are likely to be the key to the observed slower degradation and maintenance of mechanical properties.
Author Response
Please see attached.
Response to Reviewer
Reviewer 2
- The authors did add more adequate descriptions of the methods, including institutional approval of animal use. Ok.
- The authors changed added the definition of TEM to be tunneling EM in the methods. But in the Results, section 3.1, page 3, line 117, they redefine TEM to be transmission EM. I believe that this might be a mistake.
We had corrected.
- Results, section 3.1, page 3, last sentence, lines 131-132. The last sentence does not make sense in English. Plus they give references in a sentence that appears to say that they have proven something. Suggestion: re-write the last 3 sentences and put the references in the right place and clarify the rationale. This is important because they want to say this in the conclusion.
We had revised the sentence.
- Is it ion-exchanged or ion-exchange layer? Check and use only one version throughout the paper. (My preference would be ion-exchange layer, but there may be a convention in engineering.)
We unified it as ion-exchanged layer.
- Figure 12. The diagrams suggest to me that there is more Mg degradation with the coating than without because the corrosion pits are more extensive. Is this what the authors want to imply?
Figure 12 illustrated the degradation mechanism start from the micro-crack of the coating. Notably, the corrosion pits happened slower than without coating.
- Results, section 3.4, page 15, lines 198-199. The authors say: “Hydrogen evolution through degradation and the roughness of the sample surface affected the cell attachment [34-35].” This sounds like they are talking about their own findings, but then they give references. I believe that the authors are trying to say this? Hydrogen evolution. …sample surface has been shown to affect cell attachment [34-35].
Lines 198-199 is about references; the sentences are revised, and the next sentence is new findings.
- The next (last) sentence would benefit from making it clear that the authors are speculating. For example: We suggest/speculate/propose that the anodic coating improved cell attachment by improving corrosion resistance, thereby reducing hydrogen release, and increasing surface roughness compared to other metals.
Had revised.
- Response to comment 7: The small numbers in the CT images should be made into a table to support their comments about the changes in Mg degradation over time. They appear to have used the data to make statements, so please show the data in a form that can be read more easily.
Had added a table in revised.
- The authors state that they have looked at Ti and uncoated Mg screw implantation previously. This paper would benefit from a discussion in which they compare this pig experiment to their previous work, in terms of function or changes in screw degradation.
We had added description in introduction.
- Conclusion: The authors say: “the phosphate coating and ion-exchange layer were found to be the keys to slowing down the degradation and maintaining the mechanical properties.” You can’t say that these were keys because that suggests that you “proved” this. Aren’t the authors speculating? A suggested alternative statement might be: After anodic oxidation, we have characterized the existence of the phosphate coating and an ion-exchange layer, which are likely to be the key to the observed slower degradation and maintenance of mechanical properties
Had revised the sentence.
